# Discovery of a Therapeutic Agent for Glioblastoma Using a Systems Biology-Based Drug Repositioning Approach

**DOI:** 10.3390/ijms25147868

**Published:** 2024-07-18

**Authors:** Ali Kaynar, Mehmet Ozcan, Xiangyu Li, Hasan Turkez, Cheng Zhang, Mathias Uhlén, Saeed Shoaie, Adil Mardinoglu

**Affiliations:** 1Centre for Host-Microbiome Interactions, Faculty of Dentistry, Oral and Craniofacial Sciences, King’s College London, London SE1 9RT, UK; ali.kaynar@kcl.ac.uk (A.K.); saeed.shoaie@kcl.ac.uk (S.S.); 2Science for Life Laboratory, KTH-Royal Institute of Technology, SE-17121 Stockholm, Sweden; mehmet.ozcan@scilifelab.se (M.O.); xiangyu.li@scilifelab.se (X.L.); cheng.zhang@scilifelab.se (C.Z.); mathias.uhlen@scilifelab.se (M.U.); 3Department of Medical Biochemistry, Faculty of Medicine, Zonguldak Bülent Ecevit University, Zongudak TR-67100, Turkey; 4Medical Biology Department, Faculty of Medicine, Atatürk University, Erzurum TR-25240, Turkey; hasanturkez@yahoo.com

**Keywords:** glioblastoma, survival, co-expression, drug repositioning, glycosaminoglycans, extracellular matrix

## Abstract

Glioblastoma (GBM), a highly malignant tumour of the central nervous system, presents with a dire prognosis and low survival rates. The heterogeneous and recurrent nature of GBM renders current treatments relatively ineffective. In our study, we utilized an integrative systems biology approach to uncover the molecular mechanisms driving GBM progression and identify viable therapeutic drug targets for developing more effective GBM treatment strategies. Our integrative analysis revealed an elevated expression of *CHST2* in GBM tumours, designating it as an unfavourable prognostic gene in GBM, as supported by data from two independent GBM cohorts. Further, we pinpointed WZ-4002 as a potential drug candidate to modulate *CHST2* through computational drug repositioning. WZ-4002 directly targeted *EGFR* (*ERBB1*) and *ERBB2*, affecting their dimerization and influencing the activity of adjacent genes, including *CHST2*. We validated our findings by treating U-138 MG cells with WZ-4002, observing a decrease in CHST2 protein levels and a reduction in cell viability. In summary, our research suggests that the WZ-4002 drug candidate may effectively modulate *CHST2* and adjacent genes, offering a promising avenue for developing efficient treatment strategies for GBM patients.

## 1. Introduction

In recent years, significant research efforts have been directed towards unravelling the development and progression of glioblastoma (GBM), a grade 4 central nervous system (CNS) cancer noted for its diversity and aggressive nature. Although central nervous system cancers overall have a lower global incidence rate (1.7%) compared to other cancer types, GBM stands out with a relatively higher incidence rate of 3.23 per 100,000 individuals. GBM is recognized as one of the most aggressive and malignant forms of CNS cancer, leading to a high mortality rate with a median survival time of just 8 months [1,2,3,4] and an average survival time of 15–18 months [5]. GBM predominantly arises in the supratentorial region of the brain, with the highest frequency observed in the frontal lobe. It is rarely found in the cerebellum or spinal cord, where it demonstrates unique tumour behaviour [6].

Molecular studies have identified several critical biological pathways and gene expression profiles associated with the stratification and diagnosis of GBM patients [3,7,8]. Alterations in the gene expression of *PDGFRA-IDH1, EGFR* (*ERBB1*), and *NF1* can be used to identify the proneural, classical, and mesenchymal types [8]. New genetic factors involved in the progression of GBM, including the SOX transcription factor family and proteins, have been discovered using CRISPR-Cas9-based studies [9]. Histological studies have revealed that the nature of GBM is heterogeneous in terms of cell types, morphology, and oncogenic changes [10,11]. These studies may contribute to extending GBM patients’ lifespans and broaden our perspective on understanding the underlying molecular mechanism involved in the progression of GBM. However, GBM remains a significant threat due to recurrence features and its limited treatment approaches.

The current standard of care for GBM treatment involves a comprehensive, multidisciplinary approach that typically includes surgery, radiotherapy, and chemotherapy. The most commonly used chemotherapy agents target a variety of biological processes. These include DNA crosslinking (Temozolomide and its derivatives), *VEGF-A* inhibition (Bevacizumab and its derivatives), and DNA/RNA damage (Carmustine and its derivatives). Treatment regimens may vary between newly diagnosed patients and those with recurrent GBM. Recent advances in omics technologies have paved the way for personalized therapeutic strategies in GBM treatment. Innovative methods, such as antibody-based drugs, vaccines, growth factor receptor inhibitors, immune checkpoint inhibitors, and modulators of the immune system, are currently under investigation. These emerging treatment alternatives, with their disadvantages, have the potential to enhance patient quality of life and increase the overall survival rates of those with GBM [12].

The systems biology approach utilizes a diverse array of experimental and computational methods to examine complex biological systems. This includes analysing high-throughput omics data through statistical analysis, network analysis, and mathematical modelling, providing a comprehensive understanding of biological processes and interactions [13,14]. This interdisciplinary approach facilitates a thorough understanding of biological systems. Furthermore, it has been successfully applied in drug repositioning, allowing for the repurposing of existing drugs for treating different medical conditions, guided by systems biology methodologies [15,16,17]. In addition to these, a recently developed alternative treatment approach, in which glioblastoma cells are exposed to a range of tumour-treating field intensities in combination with standard chemotherapy drugs, has shown potential synergistic anti-tumour effects and received clinical approval from the Food and Drug Administration [18].

In this study, we employed a systems biology approach to identify a therapeutic target that can be used for the development of effective treatment strategies for GBM based on the integrative analysis of global gene expression profiling of GBM patients. Specifically, we developed a workflow for the discovery of the potential therapeutic agents for the treatment of GBM patients by performing (i) survival analysis by studying the associations between the survival of GBM patients and global gene expression profiling of tumours obtained from three different GBM cohorts, (ii) differentially expressed gene analysis (DEG) to reveal the altered gene expression, (iii) co-expression analysis to identify the critical gene clusters involved in the occurrence of the disease, and (iv) drug repositioning to discover the drug candidate that can modulate the key genes and clusters involved in the progression of the disease (Figure 1). Employing computational drug repositioning, we explored drug-target interactions and identified potential therapeutic agents that could modulate these target genes in GBM. Our research provides valuable insights into identifying therapeutic drug targets and discovering effective treatment agents for GBM, paving the way for future pre-clinical and clinical trials.

## 2. Results

### 2.1. Survival Analysis Identifies Prognostic Genes for GBM

In this study, we performed univariate Cox regression analysis to examine the relationship between the expression levels of protein-coding genes and patients’ survival outcomes in three different cohorts including The Cancer Genome Atlas (TCGA) cohort, CGGA_693 cohort, and CGGA_325 cohort from the Chinese Glioma Genome Atlas (CGGA). Based on our analysis, 1526, 615, and 2232 prognostic genes were identified in these three datasets, respectively (Figure 2A). We compared the results from the three datasets to ensure consistency in prognostic genes and found that they significantly overlapped (*p* < 0.05, hypergeometric test; Figure 2B). Notably, most of the overlapped prognostic genes were unfavourable, indicating that high expressions of these genes were associated with poor patients’ survival outcomes.

To confirm the reproducibility of our findings, we constructed a pool of prognostic genes by including only those that were identified as prognostic in at least two datasets based on Cox regression analysis. This resulted in a consensus pool of 471 overlapped prognostic genes, including 381 unfavourable prognostic gene pool (UPGs) and 90 favourable prognostic gene pool (FPGs) (Figure 2A). Comprehensive information regarding significant genes is available in Appendix A.

Functional enrichment analysis revealed that the UPGs were significantly enriched in the biological processes such as extracellular matrix organization, cell migration, blood vessel morphogenesis, angiogenesis, aminoglycan, and glycosaminoglycan (GAG) (Figure 3(A.1)). Similarly, we observed supportive results from Kyoto Encyclopaedia of Genes and Genomes (KEGG) pathway analysis showing the functional enrichment of biological processes, including the extracellular matrix (ECM) receptor interaction, GAG degradation, and proteoglycans. Some well-known biological pathway alterations in GBM are phosphatidylinositol-3-OH kinase (PI3K) pathway activation and growth factor signalling defects depending on receptor tyrosine kinase (RTK) activation [7,8] that are confirmed based on KEGG pathway functional enrichment analysis (Figure 3(A.2)). On the other hand, FPGs were only enriched in cytoplasmic translation processes, which are mostly related to RNA metabolism and also confirmed by KEGG pathway functional enrichment analysis. The ribosome pathway that is associated with the altered metabolism regarding the tumorigenic activity was also significantly altered (Figure 3(C.1,C.2)).

### 2.2. DEG Analysis Supports Survival Results

We performed differential expression gene (DEG) analysis by comparing the low survival group with the high survival group and observed that our results support our survival analysis results. As a result of our DEG analysis, we found that 398, 192, and 425 genes were upregulated in the TCGA dataset, CGGA_693 dataset, and CGGA_325 dataset, respectively (Figure 2(A.3)). We compared the upregulated gene groups from each dataset using a hypergeometric test and found that they were significantly overlapped (Figure 2B). In total, 60 genes were upregulated (URPGs) in at least two datasets. We did not consider downregulated genes since the TCGA and CGGA_693 datasets did not show a significant overlap in terms of downregulated genes. When we compared the UPGs using a hypergeometric test, we found that they were significantly overlapped, supporting the consistency of our DEG and survival results (Appendix A). We then performed Gene Ontology (GO) enrichment analysis using the upregulated genes to further reveal the associated functions and found that strongly exposed immune systems responses to cancer, including macrophage activation and leukocyte activation. We also found that the cell migration process was significantly enriched. This suggests that the extracellular matrix organization may be rearranged due to the cancer cell infiltration (Figure 3(B.1)). KEGG pathway functional enrichment analysis showed that only one pathway was altered: ECM-receptor interaction, which is a well-known alteration in GBM [19].

### 2.3. WGCNA Identify Mostly Connected Genes in GBM

We performed weighted gene co-expression network analysis (WGCNA) [20] to identify the modules correlating with the disease outcome, like survival. After performing WGCNA on three cohorts separately, we found 22, 11, and 25 gene modules from the TCGA, CGGA_325, and CGGA_693 datasets, respectively. Modules from the three cohorts were compared with each other to evaluate their overlap using the hypergeometric test. We observed significant overlap over the cohorts represented using the Jaccard Index (Appendix A). We used each module as a “node” to visualize significantly overlapping modules in Cytoscape. “Edge” represents a significant overlap between the two nodes. Some modules, like C2.M_1, were significantly connected with 11 modules over the three datasets (Appendix A). Functional enrichment analysis showed that this module related to important biological processes like cell morphogenesis, collagen metabolic process, cell differentiation, cytokine production angiogenesis, and extracellular matrix organization. To narrow down our results by evaluating the survival outcomes in the network centrality of modules, we compared UPGs with each module over three datasets by using the hypergeometric test (Appendix A). We obtained six modules, one from CGGA_325, two from CGGA_693, and three from TCGA, that significantly overlapped with the UPGs from three cohorts, and defined them as prognostic modules hereafter (Figure 2(D.1)). Module genes and comparative analysis outcomes from the WGCNA are presented in Appendix A.

### 2.4. Discovery of Target Genes for Effective Treatment of GBM

In GBM research, it is preferable to inhibit the target genes since the activation of genes in the brain is relatively more challenging. The UPGs were pointed out as suitable candidates since their expression increases when the patients’ survival decreases. The expression pattern of these target genes over the human body, including the human brain, was obtained from the Human Protein Atlas (HPA) portal (data from the HPA database are available in Appendix A). The goal was to identify the genes within the UPGs that are either brain-specific or brain-elevated. The main idea behind this is that when we knock out the selected gene, it should not affect the other tissues severely. This analysis revealed 13 genes that are brain-elevated compared to other tissues and 1 brain-specific gene, yielding a total of 14 target genes that were identified as suitable candidates (Table 1).

The essentiality scores of these genes were obtained from the Dependency Map (DepMap) (Appendix A), with a preference for those with scores close to −0.5. Among these 14 target genes, we focused on 4 genes, *CHST2, GLIS3, GNA12*, and *MT1F*, since the T-drug database contains drug-target information for these genes. After analysing the available data, we determined that *CHST2* is the most promising candidate, since *CHST2* is present in prognostic modules (Figure 2(D.1)) and, according to the DepMap gene effect plot (Appendix A), it may be a potentially essential gene. In addition, the expression level of *CHST2* was higher in the GBM samples than in the control group according to TCGA data, and *CHST2* was considered as an UPG in the CGGA_325 and TCGA cohorts (Figure 4). *CHST2*, known as carbohydrate sulfotransferase 2, encodes an enzyme responsible for the sulfation of complex carbohydrates. *CHST2* primarily has a role in keratan sulfate (KS) and chondroitin sulfate (CS). *CHST2* impacts heparan sulfate (HS) production by connecting with *HS6ST1*, which has a primary role in HS production. In addition to *HS6ST1, B4GALT1* also altered the gene responsible for HS production. *CHST2, CHSY1*, and *CHPF*, which are members of UPGs, are responsible for the production of CS.

On the other hand, *DRAXIN*, which plays a role in neurite formation and axon guidance, has the potential to be a target gene since it is a brain-specific gene. But *DRAXIN* was found in non-prognostic modules, and its gene effect was graphically skewed to the positive according to the DepMap gene effect plot.

### 2.5. Drug Repositioning for Treatment of the GBM

We identified potential drug candidates for the treatment of GBM based on a drug repositioning approach (Figure 5). This involved searching for target genes in the T-Drug database and integrating data on the transcriptional response to shRNA knockdown and chemical perturbation based on Connectivity Map (CMAP). We focused on *CHST2*, an enzyme involved in keratan sulphate and heparan sulphate metabolism. Our analysis followed the flowchart outlined in Figure 5. We searched available drugs that might be effective against *CHST2* in the T-Drug database. Potential drugs were identified by searching the T-Drug database for substances that have an inhibitory effect on *CHST2*. Of the top 25 drugs (Appendix A) identified in this way, 2 were selected for further analysis based on their known target genes: carbinoxamine, which targets the *HRH1* gene, and WZ-4002, which targets the *EGFR* (also known as *ERBB1*) and *ERBB2* genes (Appendix A). We found that *CHST2* and the well-known target genes of the proposed drugs, *HRH1*, and *EGFR/ERBB2*, were co-expressed in the GBM tumours as shown in iNetModels (Appendix A). When we investigated the interactive properties of *HRH1* vs. *CHST2*, there was no interaction in the brain’s cortex region in a healthy state. Similarly, we did not see any interaction between *CHST2* and *EGFR/ERBB2* in the healthy brain in the cortex region.

Hence, we found that *CHST2* and well-known target genes of WZ-4002 were significantly co-expressed in the GBM. Some significantly altered genes that have functional roles in GAG metabolism were then examined using the iNetModels database (Appendix A).

### 2.6. In Vitro Validation of Drug Candidate

The U-138 MG brain glioblastoma cancer cell line was used to evaluate the effect of the drug candidates (WZ-4002 and Carbinoxamine) on the cell viability and the protein expression of *CHST2* (Figure 6). Before determining the application doses, different concentrations of WZ-4002 and carbinoxamine were applied to U-138 MG, and cell viabilities were examined using MTT (Appendix A). According to the cell viability results, while the application of WZ-4002 at 10 μM reduced cell viability by approximately 50%, no toxic effect of carbinoxamine was observed. Therefore, concentrations of 5 and 10 μM for WZ-4002 and a higher concentration of 20 μM for carbinoxamine were applied to observe their potential effects on U-138 MG cells. After the treatments of cells with the drug candidates, the expression level of *CHST2* was measured via Western blot. We found that 5 and 10 µM WZ-4002 treatments decreased the *CHST2* expression level depending on the dose, while 20 µM Carbinoxamine was not as effective as WZ-4002 (Figure 6A). Moreover, we found that WZ-4002 slightly decreased the expression of *EGFR*. In parallel to Western blot analysis results, 10 µM WZ-4002 treatment significantly (*p* ≤ 0.001) reduced the cell viability by 50%, while Carbinoxamine treatment was not effective on the cell viability (Figure 6B). In addition, we performed the wound healing assay to demonstrate the effect of the drugs on the migration of the cells (Figure 6C). The results of the wound healing assay indicated that the treatment of 10 µM WZ-4002 completely blocked the cell migration and the treatment of 5 µM WZ-4002 slowed down. However, we observed that the wound scarring was closed in the carbinoxamine-treated cells as in the control at the end of two days. In summary, the efficacy of WZ-4002 as a drug candidate was validated by Western blot analysis as well as cell viability and wound healing assays in vitro.

## 3. Discussion

GBM is an exceptionally aggressive and lethal brain cancer, marked by disrupted cell proliferation, angiogenesis, and the invasion of adjacent tissues. This malignancy involves aberrant endothelial cell proliferation, modified vascular permeability, and the formation of new blood vessels by tumour cells [21]. In our study, we identified WZ-4002 as a promising drug candidate for GBM treatment. WZ-4002 is a third-generation *EGFR* inhibitor, belonging to the ERBB family of four transmembrane tyrosine receptor kinases: *EGFR, ERBB2, ERBB3*, and *ERBB4*. This receptor family is crucial in various cellular processes, including cell survival. The dimerization of ERBB family members triggers downstream cell signalling cascades following 3D structural alterations that allow certain agents to bind to their extracellular domain. Activating the *ERBB* family leads to the stimulation of key pathways such as the MAPK pathway, STAT3 signalling, and the PI3K–Akt pathway, which are primarily associated with cell proliferation and tumour cell survival [22,23,24]. The activation of the MAPK pathway promotes gene transcription activities controlling cell proliferation, cell migration, and angiogenesis. Conversely, the PI3K–Akt pathway activation governs vital cellular functions like anti-apoptotic activity, cell survival, and cell cycle regulation [25,26]. The ERBB family, crucial in cancer cell biology, is a target for various anticancer agents, including monoclonal antibodies, small-molecule tyrosine kinase inhibitors (TKIs), antibody-drug conjugates (ADCs), peptide-based inhibitors, immunotoxins, and heat-shock protein inhibitors. Regarding the ERBB family, a range of inhibitors has been tailored to its unique aspects. Monoclonal antibodies, such as cetuximab, panitumumab, and trastuzumab, specifically target the ERBB family to impede cancer cell proliferation to prevent *EGFR* activation by obstructing its extracellular domain from binding with growth factors. A monoclonal antibody, necitumumab, specifically approved for squamous cell lung cancer [27], similarly inhibits *EGFR* activation. In the field of small-molecule TKIs, drugs like erlotinib, gefitinib, lapatinib, and osimertinib inhibit EGFR’s tyrosine kinase domain [28]. In our initial screening, erlotinib and lapatinib were identified among the larger list (Appendix A) of candidate drugs. Both of these drugs are well-established tyrosine kinase inhibitors with known efficacy in targeting EGFR and ERBB2 pathways. Given their mechanisms of action and our preliminary findings, it is plausible that erlotinib and lapatinib could also modulate the *CHST2* and its adjacent networks. Innovative approaches such as antibody-drug conjugates (ADCs), which consist of a monoclonal antibody, and a cytotoxic drug, such as gemtuzumab ozogamicin, and peptide-based inhibitors, broaden therapeutic strategies [29,30]. Another notable example of EGFR family modulation involves heat shock protein 90 (*HSP90*), which regulates the stability of *ERBB2* [31]. *HSP90B1*, a member of the HSP90 family, plays a role in sustaining cancer cell survival by facilitating ERBB2-dependent downstream signalling pathways. The inhibition of ERBB2 signalling has been shown to disrupt cell cycle progression and hinder angiogenesis, demonstrating its potential as a therapeutic target in cancer treatment.

The ECM, one of the major altered biological processes in GBM, is a complex network of proteins and carbohydrates that surrounds and supports cells. ECM plays a critical role in the development and progression of GBM because of the crosstalk between the tumour cells and the microenvironment, which can influence the behaviour of cancer cells. Proteoglycans, essential components of the ECM, consist of a core protein containing GAG that plays a critical role in cross-talking. Altered proteoglycan content could promote tumour progression, invasion, and tumour growth [32,33]. The main components of the ECM in GBM consist of hyaluronic acid and some protein families such as tenascin, fibulin 3, and fibronectin [19]. Some members of the protein families involved in the formation and functioning of the ECM are in our prognostic gene pool, including *TNC, FN1*, and *FBLIM1*, which has changed significantly. A crucial gene in hyaluronic acid synthesis, a member of the ECM component, is *HAS2*, a member of UPGs and URPGs. ECM remodelling, regarding conditions such as cancer or brain injury, is accomplished by tightly regulated production and degradation of ECM components [34,35]. In this study, we observed that genes related to important biological events in cancer, such as cell survival, cell adhesion, cell proliferation, angiogenesis, ECM organization, and signal transduction, have been significantly altered (Figure 3). In this context, our results have been verified based on the literature.

GAGs have a crucial role in such biological functions, including ECM modulation, cell proliferation, and immune control. Altered GAGs profiles can be used as a biomarker for the early detection of several cancer types [36]. When we analysed the altered biological process and KEGG pathway results, we observed that the GAGs and GAG-related proteoglycan processes were significantly altered. The expression level of the key genes, which have a critical role in GAGs and the proteoglycan process, was increased in line with the decreased survival time. Hypothetically, we found that increased GAGs and proteoglycan levels were significantly correlated with the survival of the patients. Our analysis indicated that the genes involved in the GAGs production process, especially our target gene *CHST2*, have a prognostic profile. Several studies have shown that GAGs, including HS, KS, and CS, have an important role in the cancer process [37,38,39,40,41].

One of the UPGs and URPGs in our results is *HBEGF*, which is an important gene for the cell survival downstream signalling pathway, including MAPK/ERK, PI3K/AKT, and JAK/STAT. HBEGF binds to EGFR and activates EGFR-dependent signalling [42]. Another gene significantly altered in our study is *STC1*, which regulates the migration and invasion of GBM cells by interacting TGF-β/SMAD4 signalling cascade [43,44]. On the other hand, *CAV1* negatively regulates TGF-β/SMAD4 signalling by mediated internalization of TβRI, a TGF-β receptor [45]. In another carcinogenesis-related pathway, the Hedgehog (HH) pathway, related genes were significantly altered based on our analysis. The HH pathway altered via ligand-independent changes like *PTCH1* (downregulated) mutation and *SMO* (UPGs) mutation. The HH pathway was also altered via paracrine ligand-dependent alteration that stroma cells provoke tumour cells via *IL6, VEGF, IGF*, and *WNT* alteration, resulting in HH alteration via *PTCH1* and *SMO*, as well as reverse-paracrine ligand-dependent alteration via stroma cells, inducing tumour cell via *PTCH1* [46,47,48]. In our study, IGF binding protein family members IGFBP4, IGFBP5, and IGFBP6, which are involved in tumorigenesis, as well as genes including *WNT5A*, *DKK1, DRAXIN* and *IGFBP4*, which are involved in the WNT pathway, were significantly altered.

GBM cells cause alterations in cytokines’ expression, composed of glycoproteins and polypeptides, and interact with tumour-associated macrophages (TAM) to suppress the anti-tumour immune response. *STAT3* is active in GBM, and its expression level correlates with patients’ survival outcomes [23]. *STAT3* mediate interferon (*IFN*) and *IL6*, affecting the JAK downstream signalling in GBM. *CXCL10* expression level increases the tumour cell. GBM formation is enhanced by TNF-α secretion, which leads to *LIF* upregulation [49,50]. Cytokine activation may induce sulfotransferase, including *CHST2* overexpression in some circumstances [51].

Our research utilized various system biology approaches to investigate genes with potential roles in GBM. Among these, *EFEMP2*, identified from the UPGs (upregulated prognostic genes) pool, stands out due to its significant function in molecular cancer mechanisms. This gene is notably interconnected with *CHST2* and *ERBB2* within the C1.M_4 module, which also includes other synergistically effective genes, highlighting their common properties. *EFEMP2* also appears in the T.M_2 and C2.M_10 modules, demonstrating significant overlaps with the UPGs pool, further underscoring its prognostic relevance. *EFEMP2* (EGF-containing fibulin-like extracellular matrix protein 2) is crucial in shaping the tumour microenvironment, markedly influencing overall survival and tumour-associated macrophage dynamics in glioma [52,53]. As a fibulin family member with EGF-like calcium-binding domains, *EFEMP2* is essential for cellular architecture and signalling within tumours and may influence glycosaminoglycan metabolism. In aggressive conditions like GBM, *EFEMP2*′s high expression correlates with poorer patient outcomes. Its prognostic ability is reinforced by its association with a spectrum of macrophage activation states—from typical polarizations to an M0-like continuum—thus impacting immune responses within the tumour [52]. *EFEMP2*′s pivotal interactions with the *EGFR* play a significant role in oncogenic pathways. By binding directly to *EGFR*, it activates the EGFR/ERK1/2/c-Jun signalling pathway, promoting tumour cell proliferation and migration while also upregulating *PD-L1* expression, which aids in immune evasion [54].

Based on the literature on the above-mentioned genes and their relationships with *CHST2* within the network, we found that *CHST2* and its close neighbours could be considered a drug target for developing effective treatment strategies for GBM. Our in vitro experiments showed that targeting GAGs metabolism with the here-identified drug candidate, WZ-4002, results in decreased cell viability.

Our study, utilizing system biology approaches, including DEG analysis, co-expression network analysis, survival analysis, and in vitro validation, highlights the potential role of *CHST2* and other genes in GBM treatment. We identified promising drug candidates such as WZ-4002, an EGFR inhibitor, which targets crucial genes and their adjacent networks, including *ERBB2* and *EFEMP2*. Future studies should explore combinations of drug treatments targeting multiple network nodes to enhance therapeutic efficacy and address issues of drug resistance. Delving deeper into the role of *CHST2* in tumour biology through expanded datasets and integrative approaches could significantly improve GBM treatment strategies. Despite the promising clues, our study’s limitations stem from heterogeneity and may not fully account for the sample heterogeneity that could affect gene expression profiles. The complexity of signalling pathways and their interactions call for larger, more comprehensive datasets to accurately decipher the intricate dynamics of *CHST2* and other relevant genes in tumour biology. Future work should also consider integrating single-cell data and other omics layers to refine our understanding and overcome the current dataset limitations.

## 4. Materials and Method

### 4.1. Data Acquisition and Pre-Processing

We retrieved the global gene expression profiling data (RNA-seq dataset) of the GBM patients recruited in the TCGA project from the Genomic Data Commons (GDC) platform [55]. The dataset includes global mRNA expression levels of all protein-coding genes (Transcript Per Million (TPM) and STAR read count) and clinical information for 174 tumour samples and 5 control samples from adjacent normal tissues. We removed duplicate samples that had low purity scores according to the clinical information. In total, we screened RNA-seq data of 149 tumour samples obtained from primary solid tumour tissue samples and well-recorded clinical information, including survival time and day to the last follow-up recording. Protein-coding genes were selected depending on the embedded information in the dataset as a “protein_coding”. The dataset was downloaded on the RStudio platform using the TCGAbiolinks R package [56]. The gene expression profiles (expression data from STAR + RSEM FPKM value and STAR read counts) and clinical information of the other two datasets, CGGA_693 and CGGA_325 [57], were collected from the Chinese Glioma Genome Atlas (CGGA) (http://www.cgga.org.cn/, accessed on 9 June 2022). The RNA-seq data of normal control samples, with a sample size of 20, were also obtained from the same database (http://www.cgga.org.cn/, accessed on 9 January 2022) by downloading the CGGA_RNAseq_Control dataset. We screened all samples of GBM and kept only primary tumour solid tissue samples and well-recorded clinical information, including survival time. After pre-processing, we included 133 and 85 samples for CGGA_693 and CGGA_325 in our analysis, respectively (detailed sample information can be found in Appendix A). In each cohort, genes with a mean expression value < 1 were considered as unexpressed, and they were excluded from the data to reduce the noise. After removing the lowly expressed genes, 14,552, 13,298, and 13,121 genes remained for the TCGA dataset, CGGA_693 dataset, and CGGA_325 dataset, respectively (Figure 1A). The sample distribution of the datasets is presented in Appendix A. The analysis was performed using RStudio (R version 4.1.0).

### 4.2. Survival Analysis

The univariate Cox regression model was used to evaluate the relationship between the gene expression levels and patients’ survival outcomes, and the hazard ratio of each gene was calculated. Genes with *p*-value < 0.05 were identified as prognostic genes based on survival analysis. If a gene’s high expression was correlated with the poor survival outcomes of GBM patients, this gene was defined as an unfavourable prognostic gene; otherwise, it was defined as a favourable prognostic gene. Favourable and unfavourable genes were determined by performing the same analysis for the three cohorts independently. In addition, a pairwise comparison was performed for the prognostic genes by hypergeometric analysis. Unfavourable or favourable genes that were significant in at least two datasets were combined to form an unfavourable prognostic gene pool (UPGs) or favourable prognostic gene pool (FPGs), respectively (Appendix A). All analyses were performed using the R package “survival”.

### 4.3. DEG Analysis

In the DEG analysis, we used the gene count values of each sample selected from each dataset. In the survival analysis, count values in the selected samples were used, and lowly expressed genes were removed from the data. Finally, the obtained data were used in the DEG analysis. The samples were sorted according to the number of survival days, and two groups were formed, namely genes associated with high survival and low survival. The quantile degrees of the data were considered when creating the high survival and low survival groups: the first quantile was chosen as 0.33, and the second quantile as 0.66. Samples with survival values lower than the first quantile were classified as low survival. Conversely, samples with a survival value greater than the second quantile were grouped as high survival. The high survival group was chosen as the reference and compared with the low survival group. The same procedure was performed for the three datasets, separately. Results with *p*-values < 0.01 were considered as significant. DEG analysis was performed using the R package “DESeq2” [58] in the R platform [59].

### 4.4. Co-Expression Network Analysis

In this study, we used weighted gene co-expression network analysis (WGCNA) by employing a series of correlations to identify the sets of co-expressed genes in each dataset. We used count data as input from the GBM samples. We performed Pearson’s correlation matrices for all pairwise genes and constructed a weighted adjacency matrix using a power function that incorporated both the Pearson correlation and adjacency between each gene pair.

Before constructing the adjacency matrix, we removed some genes with low standard deviation. Accordingly, the cut-off value of the dataset to determine the low standard deviation was 0.25,0.35, and 0.3 for the datasets TCGA, CGGA_693, and CGGA_325, respectively. The power parameter, which controlled the strength of the correlations, was optimized for each cohort to achieve a scale-free topology fit (R^2^), and the cut-off was selected as 0.85. We set the power parameters as 12, 13, and 11 for TCGA, CGG_325, and CGGA_693, respectively (Appendix A). We then converted the adjacency matrix into a topological overlap matrix (TOM) to determine the connectivity of each gene in the network. Following this, we generated a TOM dissimilarity matrix. Modules were identified using the dynamic tree-cut selecting method as a hybrid, following numeric labels converted into colours. Eigengenes were calculated, and modules were merged with similar expression profiles. To summarise, the hierarchical clustering analysis was performed on the TOM to identify the groups of genes with a similar expression pattern. A minimum group size of 20 genes was used as a threshold for the dendrogram (Appendix A).

### 4.5. Functional Enrichment Analysis

GO term lists were retrieved from Ensembl Biomart from Ensembl Release 106 using the “getBM” function. We obtained the GO terms using the “enrichGO” function of “clusterProfiler”, an R package, to analyse the functional enrichment of the DEGs. GO terms with Benjamini–Hochberg-adjusted *p*-values <0.05 were determined to be statistically significant. We obtained the KEGG pathway terms using the “enrichKEGG” function of “clusterProfiler”, an R package, for pathway enrichment analysis of DEGs. KEGG pathway terms with Benjamini–Hochberg-adjusted *p*-values <0.05 were determined as statistically significant.

### 4.6. Drug Target Identification

Since unfavourable genes are associated with the poor survival of GBM patients, knockout of these genes could theoretically kill cancer cells or reduce proliferation. Further, genes showing high expression in the brain based on the data in the Human Protein Atlas (20 November 2022) were compared with UPGs. As a result, 13 overlapped genes were identified as brain elevated, and one was identified as brain-specific (Table 1). These genes were selected as a candidate drug target pool. In addition, essential scores of these genes were searched from the Dependency Map (DepMap) portal (https://depmap.org/portal/, accessed on 20 November 2022) to investigate the perturbation effects. A lower gene score indicates that this gene has more potent effects on tumour cell proliferation after CRISPR-Cas9 or RNA interference knockout. In most cases, a gene with an essential score less than −0.5 means that the cells may grow more slowly when the gene is knocked out.

### 4.7. Drug Repositioning

In our previous study, we proposed a drug repositioning method based on the similarity analysis of the transcriptomics signature profiles of human cells treated with drugs and genetic perturbations. We hypothesised that the inhibitory effect of a drug on the expression of a target gene can be inferred if the drug causes a perturbation in the gene expression landscape of tumour cells (in this case), similar to the effect of shRNA-mediated knockdown of the target gene. The approach for drug repositioning comprises four steps: construction of the drug-shRNA matrix, optimization of the matrix, identification of the top 1% of drug candidates, and selection of the most potent drugs for each target gene [15]. Based on this drug repositioning approach, we constructed a web tool named T-Drug (http://drug-reposition.com, accessed on 18 December 2022), in which users can freely query a specific gene or drug. T-Drug provides a ranked list of predicted gene-drug associations based on the confidence scores. Based on the results, we sorted the candidate drugs in descending order in the number of hit cells, short for “Num Hit Cells”. The first 25 drugs were selected as candidate drugs. Then, drugs and their well-known target gene information (compoundinfo_beta.txt) were downloaded from CMAP to evaluate the relationship between the candidate genes and selected drugs. Next, we examined the interaction between the *CHST2* and the well-known target genes using the iNetModels 2.0 [60] and investigated how candidate drugs can potentially modulate it and its neighbouring genes.

### 4.8. Cell Culture and Drug Treatment

The U-138 MG cells were maintained in Dulbecco’s Modified Eagle Medium (D0822, Sigma Aldrich, St. Louis, MO, USA) supplemented with 10% fetal bovine serum (FBS) and 1% penicillin/streptomycin (P/S) solution in a humidified incubator with 5% CO_2_ at 37 °C. For the drug treatments, 2.5 × 10^5^ cells were seeded to per well in a 6-well plate. After 24 h of cell seeding, cells were treated with the WZ-4002 and Carbinoxamine (HY-12026, HY-B1589A, MedChem Tronica Ab, Stockholm, Sweden) dissolved in 0.1% Dimethyl sulfoxide (DMSO, 41639, Sigma-Aldrich) for two days.

### 4.9. Western Blot Analysis

After the drug treatments for 2 days, the cells were washed with phosphate-buffered saline (PBS) and lysed with CelLytic M (C2978, Sigma-Aldrich) lysis buffer containing protease inhibitors (11836170001, Roche, Buchs, Switzerland). The cell lysates were centrifuged at 12,000 rpm and +4 °C for 10 min, and then supernatants were collected. Protein Assay Dye Reagent (5000006, Bio-Rad, Hercules, CA, USA) was used for the determination of protein concentration. The absorbance of the proteins was measured spectrophotometrically at 595 nm by a microplate reader (Hidex Sense Beta Plus). Protein electrophoresis was performed using Mini-PROTEAN^®^ TGX^TM^ Precast Gels (4561086, Bio-Rad), and then the proteins were transferred to a Trans-Blot Turbo Mini 0.2 um PVDF Transfer Packs membrane (1704158, Bio-Rad) using the Trans-Blot^®^ Turbo^TM^ Transfer System (Bio-Rad). The membrane was blocked with 5% skim milk for 30 min at 4 °C with gentle rocking. After blocking, the membrane was treated with the primary antibodies: Anti-CHST2 (PA5-69408, Invitrogen, Carlsbad, CA, USA), Anti-EGFR (ab52894, Abcam, Cambridge, UK) and Anti-Beta actin (ab8227, Abcam) as a loading control, overnight at 4 °C on a rocking platform. After the treatment of primary antibodies, the membrane was washed three times with TBS-T buffer (A09-7500-100, Medicago, Uppsala, Sweden). Then, Goat anti-Rabbit IgG-HRP (ab205718, Abcam) was treated as the secondary antibody for 30 min at 4 °C with gentle rocking. The protein bands on the membrane were revealed using enhanced chemiluminescence substrate (WBLUF0500, Merck, Rahway, NJ, USA) and detected with ImageQuant™ LAS 500 (GE Healthcare, Medison, WI, USA).

### 4.10. Cell Viability Assay

The cytotoxic effects of the drugs on the U-138 MG cell line were tested by MTT (Thiazolyl Blue Tetrazolium Bromide) analysis. Therein, 1 × 10^4^ cells were seeded to per well in a 96-well plate in 100 µL of DMEM, 10% FBS and 1% P/S. After 24 h of cell seeding, different concentrations of the drugs were treated on the cells for two days. After the incubation period, 5 mg/mL MTT solution in PBS (10 µL) was added to each well. After 2 h, MTT solutions were removed from the wells. Then, 100 μL of DMSO was added and mixed to dissolve the formazan crystals. Cell viability was analysed by measuring the absorbance of the dissolved formazan at a wavelength of 570 nm.

### 4.11. Wound Healing Assay

To investigate the effects of the selected candidate drugs on cell migration using U-138 MG cells, a wound healing assay was performed. Therein, 1 × 10^5^ cells were seeded into 24-well plates for each well and grown until 95–100% confluent. After the medium was removed, a linear wound was created using a 200 μL pipette tip, and the image was recorded at the time point of day 0. Cells were washed with PBS, and the drugs were treated in DMEM containing 2% FBS. Then, images of the wound surfaces were recorded after one and two days by the ZOE Fluorescent Cell Imager (Bio-Rad, Hercules, CA, USA).

## 5. Conclusions

In conclusion, our study introduces a drug repositioning method grounded in mRNA expression profiling, aimed at developing alternative therapeutic strategies for GBM. This approach facilitated the identification of *CHST2* as a gene with altered expression linked to patient survival outcomes. Subsequently, we pinpointed disease-related hub genes through network topology and co-expression network analysis. The impact of drug perturbation on these selected genes was assessed through in vitro experiments, leading to the identification of WZ-4002 as an effective drug for modulating *CHST2*. Our findings suggest that this methodology offers a promising, cost-effective avenue for expediting drug development.

## Figures and Tables

**Figure 1 ijms-25-07868-f001:**
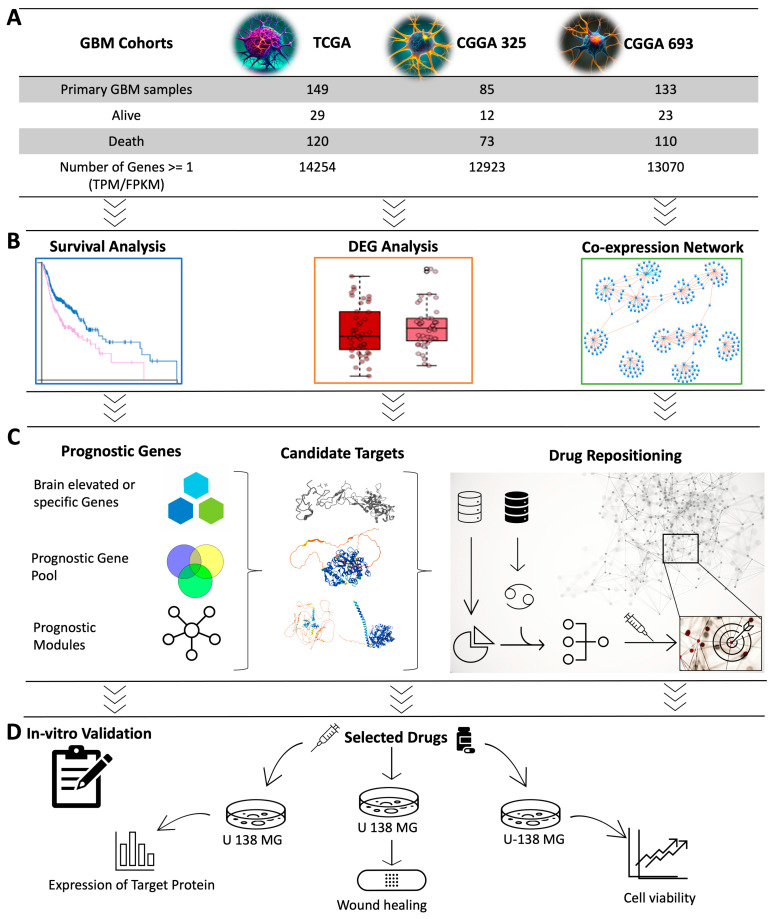
Overview of study design and methodology. (**A**) Data from three different GBM cohorts were included in the study. (**B**) Representation of computational systems biology approaches used in the analysis was visualized. Cox survival analysis was used to identify the prognostic genes whose expression levels indicated the GBM patients’ survival outcomes. Weighted gene co-expression network analysis (WGCNA) was performed to identify the key gene modules related to patients’ prognoses and to investigate the centrality of genes in these modules. DEG analysis was performed to identify the genes associated with the survival analysis and WGCNA. (**C**) The sequence from the prognostic gene pool to the drug repositioning process is illustrated. Prognostic genes were narrowed to discover the candidate target genes suitable for drug repositioning. Drug repositioning was performed to identify promising drug candidates for modulating the target genes and their neighbouring genes in the gene clusters. (**D**) In vitro validation was performed to test the effect of the drug candidate.

**Figure 2 ijms-25-07868-f002:**
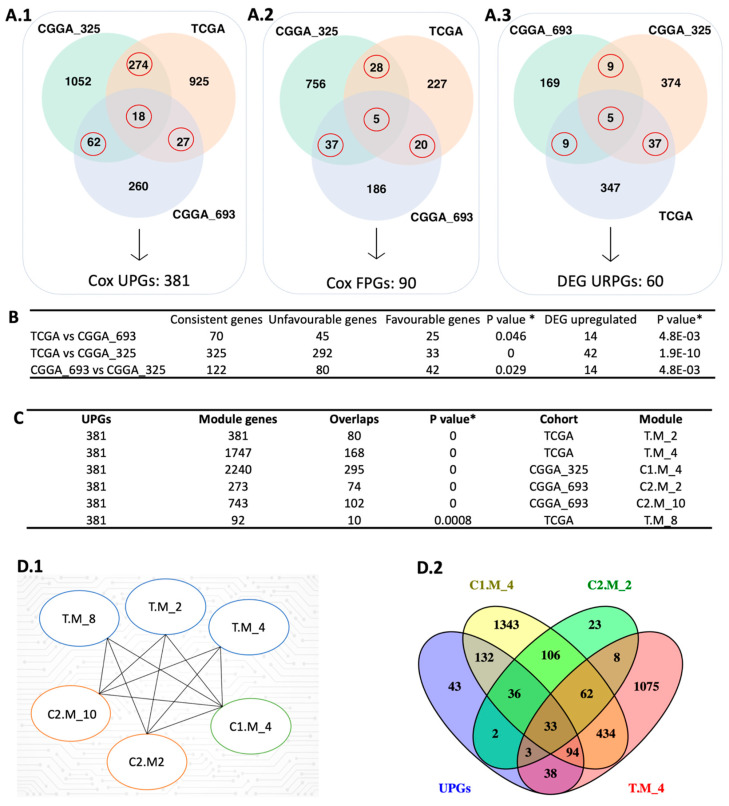
Integrative systems biology approaches for the identification of prognostic gene markers. (**A**) The red circle represents the intersection between the Cox-survival prognostic analysis. The number of unfavourable genes (**A.1**), favourable genes (**A.2**), and upregulated prognostic genes (**A.3**) are shown in three different datasets. (**B**) The hypergeometric test was performed to compare each dataset’s Cox and DEG results. (**C**) The figure shows the comparison of UPGs and modules derived from WGCNA using the hypergeometric test. (**D**) The figure represents the significantly overlapped modules with the UPGs (**D.1**) and the number of intersection genes between selected modules (**D.2**). In the module name, T, C1, and C2 affix represent TCGA, CGGA_325, and CGGA_693, respectively. * Hypergeometric test: *p*-value ≤ 0.05.

**Figure 3 ijms-25-07868-f003:**
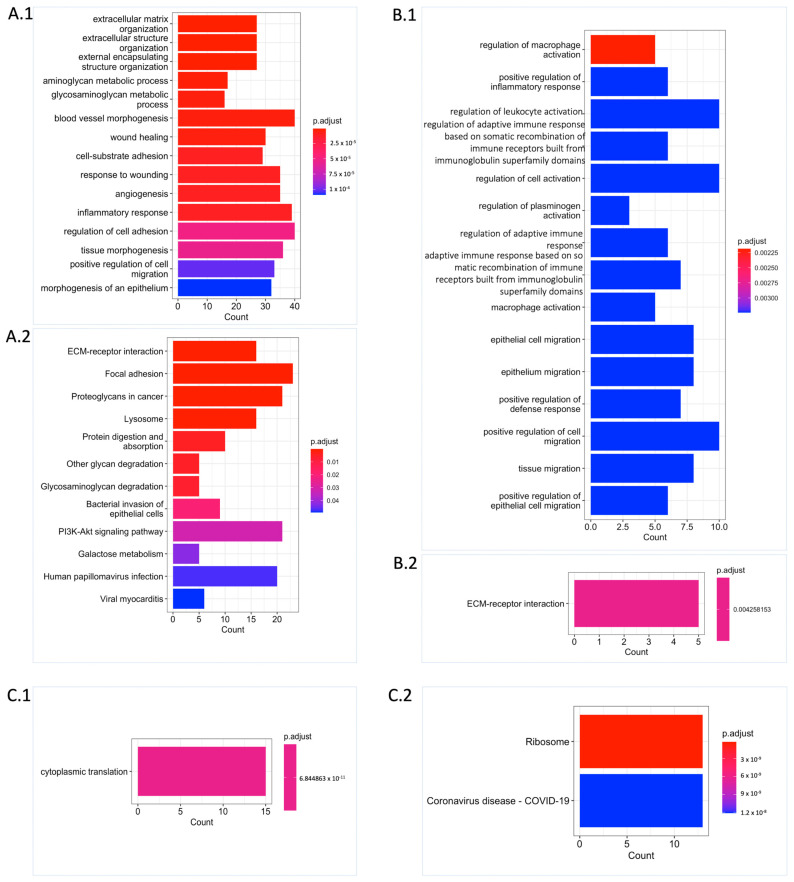
Functional enrichment analysis unveiling significant biological process and pathway alterations. (**A**) Significantly altered Gene Ontology (GO) biological processes (**A.1**) and KEGG pathways (**A.2**) for UPGs are shown. (**B**) Significantly altered GO biological processes (**B.1**) and KEGG pathway (**B.2**) for URPGs are shown. (**C**) Significantly altered GO biological process (**C.1**) and KEGG pathways (**C.2**) for FPGs are shown. Count: This represents the number of genes from the relevant gene pool that are included in the corresponding term. For each category, the cut-off value for significantly altered terms was set at p.adjust ≤ 0.05 obtained with Benjamini–Hochberg adjustment.

**Figure 4 ijms-25-07868-f004:**
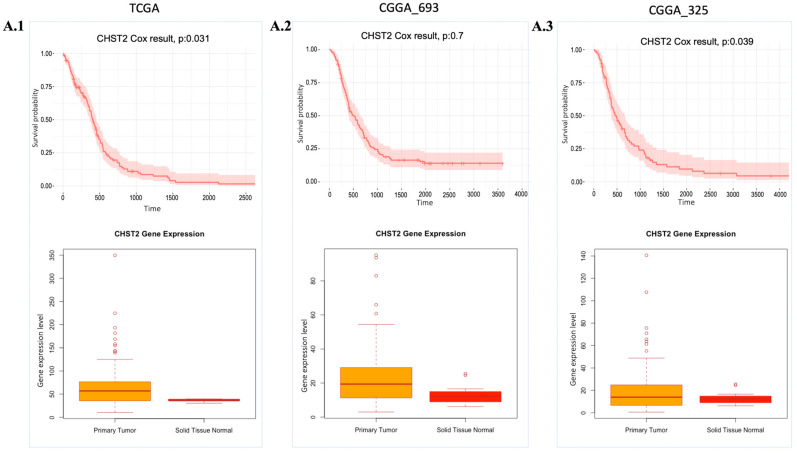
Expression and cell survival profile of *CHST2*. Cox-survival results and gene expression (Transcript Per Million (TPM) value) profile of *CHST2* comparing health conditions are presented. (**A.1**–**A.3**): *CHST2* gene expression levels and Cox survival analysis results from the TCGA, CGGA_693, and CGGA_325 cohorts, respectively. *p*: *p*-value.

**Figure 5 ijms-25-07868-f005:**
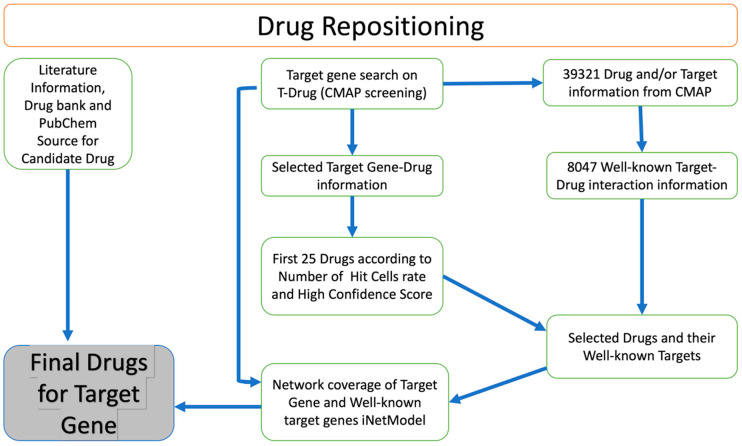
Overview of drug repositioning strategy. Drug repositioning strategy used in this study.

**Figure 6 ijms-25-07868-f006:**
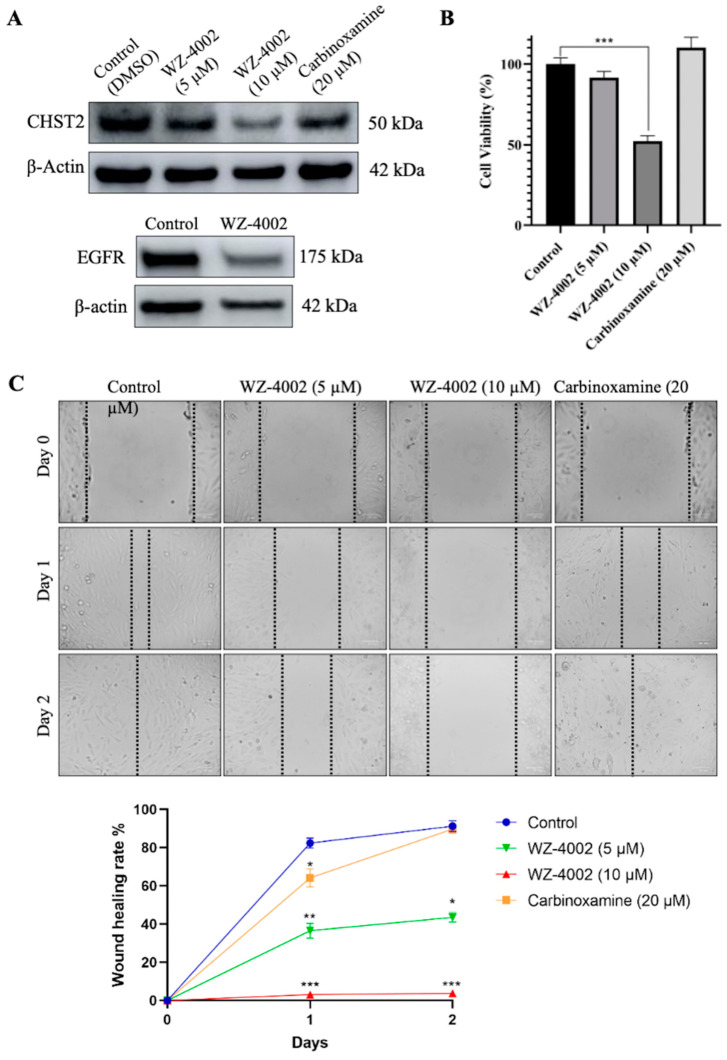
Efficacy of candidate drugs on GBM cell line: in vitro validation. The effect of WZ-4002 and Carbinoxamine treatment on the cell viability/migration as well as the target protein *CHST2* is presented. (**A**) The expression levels of target protein (*CHST2*) after treatment of WZ-4002 and Carbinoxamine in U-138 MG cells for two days are presented. The effect of WZ-4002 (10 µM) on *EGFR* expression is shown. β-actin was used as a loading control. (**B**) The evaluation of the U-138 MG cell viability after two days of the WZ-4002 and Carbinoxamine treatments is presented. (**C**) Images from wound healing experiments at different time points. Scale bar = 100 μm. Cell viability and wound healing rates are presented as means ± SD from triplicate measurement and * *p* ≤ 0.05, ** *p* ≤ 0.01, *** *p* ≤ 0.001.

**Table 1 ijms-25-07868-t001:** Overview of candidate genes. Information about candidate target genes. * Elevated in brain tissue or cerebrospinal fluid, ** brain-specific.

Genes	Short Information
*ARRDC4 **	Arrestin domain containing 4 is involved in the regulation of cell growth and survival.
*CHST2 **	Carbohydrate sulfotransferase 2 is involved in the synthesis of sulfated proteoglycans and plays a role in the extracellular matrix.
*CHST6 **	Carbohydrate sulfotransferase 6 is involved in the synthesis of sulfated proteoglycans and plays a role in the extracellular matrix.
*CLU **	Clusterin is involved in the extracellular matrix and it is important for cell adhesion and migration.
*DIRAS3 **	DIRAS family GTPase 3 is involved in the regulation of cell growth and survival.
*EN1 **	Engrailed homeobox 1 is involved in the development of the nervous system and plays a role in axon guidance.
*GLIS3 **	GLIS family zinc finger 3 is involved in the regulation of gene expression and plays a role in the development of the kidney.
*GNA12 **	G protein subunit alpha 12 is involved in the regulation of cell growth and survival.
*IBSP **	Integrin-binding sialoprotein is involved in the extracellular matrix and is important for cell adhesion and migration.
*LCTL **	Lactase-like, the function of which is to hydrolyse glycosidic bonds and involved in sensory transduction.
*LZTS1 **	Leucine zipper, putative tumour suppressor 1, is involved in the regulation of cell growth and survival.
*MT1F **	Metallothionein 1F is involved in the regulation of metal ions and plays a role in the response to oxidative stress.
*SCARA3 **	Scavenger receptor class A member 3 is involved in the recognition and clearance of damaged cells and plays a role in the immune system.
*DRAXIN ***	Dorsal inhibitory axon guidance protein is involved in the development of the nervous system and plays a role in axon guidance.

## Data Availability

Data are contained within the article and Appendix A. The code is available at https://github.com/alikaynar-kcl/Glioblastoma_P1_AK.git (accessed on 2 May 2024).

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
