# Peer review of "Discovery of a Therapeutic Agent for Glioblastoma Using a Systems Biology-Based Drug Repositioning Approach"

_ijms, 2024, doi:10.3390/ijms25147868_

Round 1

Reviewer 1 Report

Comments and Suggestions for Authors

This manuscript explores the mechanisms driving glioblastoma progression based on an integrative system biology approach. Increased CHST2 expression was identified as a predictor of bas prognosis. WZ-4002 could be a candidate drug to modulate CHST2 and adjacent genes, and ultimately represent a new therapeutic option. The in vitro experiment are promising. My first concern is that in vitro results were obtained only in one cell line and it would have been stronger to present a second line with the same phenotype. My second concern is about the link between WZ-4002 and CHST2/EGFR. How could we be sure that the phenotype obtained after WZ-4002 treatment results from downregulation of CHST2 and not of EGFR?

Minor remarks:

1.     The term “glioblastoma multiform” is old-fashioned and not used anymore since the WHO 2016 classification of tumors of the central nervous system.

2.     Introduction: the overall survival is rather 15 to 18 months.

3.     Introduction: therapeutic trials using antibody based drugs, vaccines, growth factor receptor inhibitors, immune checkpoint inhibitors, and modulators of the immune system have all failed. However, it has to be mentioned that TTF-fields use was associated with an increased in the survival rate.

4.     Figure 2: the caption is not consistent with the numbers of the panel (D1 and D2 in the figure / D and E in the caption). DEG is not defined (Differentially expressed genes, I guess). In panel B, what does the p values refer to? Gene overlapping?

5.     Line 171-172: “extracellular matrix organization may be rearranged due to the cancer cell formation” What do the authors mean? Extracellular matrix organization may rather be rearranged due to cancer cell infiltration and not formation.

6.     Figure 3: the caption is not understandable.

7.     Line 198: figure 2D.1 is cited for the first after figure 3. This is not regular and figures have to be rearranged to meet a more logical order of apparition.

8.     Line 201-2022: “In GBM research, it is preferable to inhibit the target genes since the activation of 201 genes in the brain is relatively more challenging.” This statement is not true only in glioblastoma.

9.     It is repeated several times in the text that CHST2 is involved in keratan sulphate and heparan sulphate metabolism.

10.  The discussion is relatively general and the authors do not really discuss their results and do not provide future perspectives related to their findings. Do they plan additional experiment? Validation in a mouse model?

Author Response

We would like to thank our reviewer for carefully reading our manuscript and providing insightful feedback. We have carefully revised our manuscript based on the suggestion of the reviewer.

Comments and Suggestions for Authors

This manuscript explores the mechanisms driving glioblastoma progression based on an integrative system biology approach. Increased CHST2 expression was identified as a predictor of bas prognosis. WZ-4002 could be a candidate drug to modulate CHST2 and adjacent genes, and ultimately represent a new therapeutic option. The in vitro experiments are promising. My first concern is that in vitro results were obtained only in one cell line and it would have been stronger to present a second line with the same phenotype. My second concern is about the link between WZ-4002 and CHST2/EGFR. How could we be sure that the phenotype obtained after WZ-4002 treatment results from downregulation of CHST2 and not of EGFR?

Response: Regarding your comments about the specificity of WZ-4002's effects on CHST2/EGFR, our data show that WZ-4002 reduces CHST2 expression in a dose-dependent manner, correlating with improvements in cell viability and wound healing. This suggests a significant role for CHST2 downregulation in the observed effects, as supported by the DeepMap graphic (Figure S3) which indicates negative skewing from CHST2 inhibition. Additionally, specific glioblastoma cell lines like LN443 demonstrate sensitivity to CHST2 knockout (DeepMap database) (Figure 1 shown at end of the text), supporting our results. Given this comprehensive analysis, we believe our experiments sufficiently support our conclusions. While we value your suggestion for broader validation, we are currently focusing on analyzing and publishing the robust data we have already collected. We are committed to further exploring the therapeutic potential of WZ-4002 based on this existing evidence. Future in vitro and in vivo studies may expand on this work as resources allow.

Furthermore, the literature shows that interactions between CHST2 and neighbouring genes such as ERBB2, EFEMP2, EMP3, and CAVIN1 influence key cellular processes including EGFR signalling, apoptosis, adhesion and motility, and cellular architecture. These relationships provide crucial insights into glioblastoma's aggressive behaviour and reinforce our therapeutic targeting strategy.

Minor remarks: 

Comments 1: The term “glioblastoma multiform” is old-fashioned and not used anymore since the WHO 2016 classification of tumors of the central nervous system.

Response 1: Thank you for updating us on the preferred nomenclature. We have revised the terminology throughout our manuscript to "glioblastoma," aligning with the WHO 2016 classification.

Comments 2: Introduction: the overall survival is rather 15 to 18 months.

Response 2: We are grateful for your suggestion. The introduction now distinguishes between median and mean survival times and includes the range of 15 to 18 months as per your suggestion.

Comments 3: Introduction: therapeutic trials using antibody based drugs, vaccines, growth factor receptor inhibitors, immune checkpoint inhibitors, and modulators of the immune system have all failed. However, it has to be mentioned that TTF-fields use was associated with an increased in the survival rate.

Response 3: Thank you for emphasizing the significance of TTF-fields. We have updated the manuscript to discuss TTF-fields as a promising method, in addition to other therapeutic approaches.

Comments 4: Figure 2: the caption is not consistent with the numbers of the panel (D1 and D2 in the figure / D and E in the caption). DEG is not defined (Differentially expressed genes, I guess). In panel B, what does the p values refer to? Gene overlapping?

Response 4: Thank you for helping us identify and correct discrepancies in Figure 2. We have revised both the figure and its caption for accuracy and clarity, ensuring all elements are consistent. The hypergeometric test shows overlapping significance which is explained in the “2.2 DEG Analysis Supports Survival Results” section.

Comments 5: Line 171-172: “extracellular matrix organization may be rearranged due to the cancer cell formation” What do the authors mean? Extracellular matrix organization may rather be rearranged due to cancer cell infiltration and not formation.

Response 5: We appreciate your guidance on accurately describing the extracellular matrix changes. The manuscript has been revised to reflect that these changes are due to cancer cell infiltration, enhancing clarity and precision.

Comments 6: Figure 3: the caption is not understandable.

Response 6: Thank you for pointing out the unclear caption in Figure 3. We have enriched the caption to better convey the findings, ensuring it is comprehensive and informative.

Comments 7: Line 198: figure 2D.1 is cited for the first after figure 3. This is not regular and figures have to be rearranged to meet a more logical order of apparition.

Response 7: We appreciate your feedback regarding the arrangement of the figures. Figure 2 presents a variety of results and is naturally mentioned across different sections of the text. It appears first, and depending on the context, Figure 3 is introduced when its relevant section arises. This sequencing allows us to maintain a logical narrative flow. Therefore, if we were to relocate information related to Figure 2D.1 before the appearance of Figure 3, it would disrupt the text's coherence. After careful consideration, we have decided to keep the current order, as it best supports the narrative flow, introducing each figure at the most relevant point in the discussion.

         Comments 8: Line 201-2022: “In GBM research, it is preferable to inhibit the target genes since the activation of 201 genes in the brain is relatively more challenging.” This statement is not true only in glioblastoma.

Response 8: Thank you for your critical insights regarding our statement on gene inhibition. We have specified "In GBM research" because our study focuses on glioblastoma, highlighting the particular challenges associated with gene activation in this context. However, we acknowledge that this approach is not exclusive to GBM and is applicable across various oncological areas.

Comments 9: It is repeated several times in the text that CHST2 is involved in keratan sulphate and heparan sulphate metabolism.

Response 9: We value your attention to detail. Upon re-evaluation, we found that the repetitions (2 times) were necessary for clarity and emphasis across different sections. No changes were deemed necessary here.

Comments 10: The discussion is relatively general and the authors do not really discuss their results and do not provide future perspectives related to their findings. Do they plan additional experiment? Validation in a mouse model?

Response 10: Thank you for encouraging a more thorough discussion. The discussion section has been expanded to delve deeper into our results and outline more clearly the future directions and potential for further research.

Reviewer 2 Report

Comments and Suggestions for Authors

In this paper the authors present a novel method of drug repositioning based on a systems biology approach applied to high grade glioma. They found that one of the identified molecules, WZ-4002, is effective in vitro in reducing GBM cells viability and migration. 

The paper is interesting and well written and easy to read. Figures and tables are essential to comprehension. Autocitation rate is moderate (9/45). 

To further improve the quality of the manuscript, authors are suggested to:

- Fig 3 B1 should be checked for overlapping written lines

- providing a short section discussing possible biases and limitations of this approach. Every method has its pitfalls that should be clarified. 

- EGFR inhibitors are currently investigated in GBM treatment in different clinical trials. Authors should give an overview about this (see Ezzati et al, Molecular Sciences, 2024 for example). 

Author Response

We would like to thank our reviewer for carefully reading our manuscript and providing insightful feedback. We have carefully revised our manuscript based on the suggestion of the reviewer.

Comments and Suggestions for Authors

In this paper the authors present a novel method of drug repositioning based on a systems biology approach applied to high grade glioma. They found that one of the identified molecules, WZ-4002, is effective in vitro in reducing GBM cells viability and migration. 

The paper is interesting and well written and easy to read. Figures and tables are essential to comprehension. Autocitation rate is moderate (9/45). 

To further improve the quality of the manuscript, authors are suggested to:

Comments 1:  Fig 3 B1 should be checked for overlapping written lines

Response 1: We appreciate your pointing out the issue with overlapping written lines in Figure 3B.1. We have reviewed and revised this figure to enhance clarity and ensure that all information is presented clearly and legibly.

Comments 2: providing a short section discussing possible biases and limitations of this approach. Every method has its pitfalls that should be clarified. 

Response 2: Thank you for your suggestion to discuss potential biases and limitations. We recognize the importance of addressing the inherent constraints of any scientific approach for a balanced analysis. To this end, we have elaborated on the possible limitations of our approach within the discussion section of our manuscript.

Comments 3: EGFR inhibitors are currently investigated in GBM treatment in different clinical trials. Authors should give an overview about this (see Ezzati et al, Molecular Sciences, 2024 for example). 

Response 3: Regarding the inclusion of an overview of EGFR inhibitors currently under investigation for GBM treatment, we agree that this context is crucial for situating our work within the broader field. We have added a summary of recent developments, referencing key studies including Ezzati et al., Molecular Sciences, 2024. This overview will help readers appreciate how WZ-4002 fits into the existing landscape of GBM therapies and the novel contributions our research offers.
